# Evolution of Nucleophilic High-molecular-weight Organic Compounds in Ambient Aerosols

**Chen He[1], Hanxiong Che[2], Zier Bao[2], Yiliang Liu[2], Qing Li[2], Miao Hu[3], Jiawei Zhou[2], Shumin Zhang[4], Xiaojiang Yao[2], Quan Shi[1], Chunmao Chen[1], Yan Han[2], Lingshuo Meng[2], Xin Long[2], Fumo Yang[5] and Yang Chen[2*]**

[1]State Key Laboratory of Heavy Oil Processing, China University of Petroleum, Beijing 102249, China.

[2]Research Center for Atmospheric Environment, Chongqing Institute of Green and Intelligent Technology, Chinese Academy of Sciences, Chongqing 400714, China.

[3] CNOOC Institute of Chemicals & Advanced Materials, Beijing 102200, China.

[4] Institute of Basic Medicine and Forensic Medicine, North Sichuan Medical College, Nanchong 637000, Sichuan, China.

[5] Department of Environmental Science and Engineering, College of Architecture and Environment, Sichuan University, Chengdu 610065, China

*Correspondence to*: Yang Chen (chenyang@cigit.ac.cn)

**Abstract.** Nucleophilic high-molecular-weight organic compounds (HMWOC) are sensitive to proton ($H^+$) in FT-ICR MS analysis. A comprehensive evaluation of diurnal evolution of nucleophilic HMWOC was performed. HMWOCs aged significantly in daily cycles, accompanied by functionality shifts, particularly oxygenated and reduced nitrogen (CHON and CHN) and oxygenated organics. The intensities of HMW oxygenated compounds increased during daytime and nighttime. The daytime evolution produced more nitrogen-containing compounds with carboxylic groups (–COOH) homologs with molecular weights greater than 300, while the nighttime evolution produced mostly small CHON compounds (molecular weights <300). During evolution, nighttime CHON removals were also observed; meanwhile, carboxylation was also identified in CHON groups. The daytime evolution produced significantly more reduced nitrogen-containing compounds and a day- and nighttime increase in CHN compounds with five members was also observed. This study can provide insights into the aging of less polar organic aerosols.

Keywords: high molecular weight organic aerosol; evolution; functionality

## 1 Introduction

Organic aerosol (OA) is a key component of atmospheric aerosols, accounting for up to 90% of submicron aerosols (Zhang et al., 2015; Tao et al., 2017; Zhang et al., 2007). OA affects solar radiation forcing, fog-cloud process, and human health (Pöschl, 2005; Creamean et al., 2014). OA can evolve due to semi-volatile OA vapor oxidation from primary OA (Jimenez et al., 2009), or heterogeneous reactions (Ervens et al., 2011).

OAs can undergo evolution during their lifetimes, resulting in changes in physicochemical properties (Ditto et al., 2021). The aged biomass burning emitted aerosols can be important sources of brown carbon (BrC) (Hodshire et al., 2019). During evolution, important components of HULIS, such as organosulfate and organonitrates, have been identified and observed in both laboratory and field measurements (Hallquist et al., 2009; Liggio and Li, 2006; Li et al., 2017b; Liu et al., 2015). The heterogeneous reactions, such as acid–base reactions of amines/ ammonia with organic acids and carbonyls, can generate nitrogen-containing organics (Ervens et al., 2011; Zhang et al., 2015; George et al., 2015). The reduced nitrogen-containing compounds can be from the reactions of amines/ammonia and carbonyls (Zarzana et al., 2012; Liu et al., 2015). These processings were important during long-range transport and severe haze formation due to stagnant air conditions in China (Li et al., 2017a).

The evolution of high-molecular-weight organic compounds (HMWOCs, M.W. larger than 200) is still an unsolved issue. For example, the overall oxidative state escalating was commonly observed, but how the oxidation and other processing occurred in HMWOCs. Most recently, online aerosol mass spectrometry has been widely used in evaluating OA, but the loss of molecular information resulted in difficulties in investigating the aerosol HMWOCs (Zhang et al., 2011; Ditto et al., 2018). HMWOCs, commonly containing elements such as C, H, O, N, and

S, can reach up to 1000 da in m.w. (Ervens et al., 2011). As a class of critical light-absorbing components and precipitation participants, the environmental behavior of HMWOC is vital to investigate to fully understand the impact of OA(Yun et al., 2019) (Bandowe and Meusel, 2017).

Fourier Transform Ion Cyclotron Resonance Mass Spectrometry (FT-ICR MS) is capable of extremely high mass accuracy and resolution for chemical analysis. It has been utilized extensively for characterizing complex organic mixtures of atmospheric OA (Xie et al., 2020; Jiang et al., 2014; Bianco et al., 2018). The method has been used to study HUMIC-like substances, typically biomass burning and coal combustion aerosols (Li et al., 2022; Song et al., 2022; Tang et al., 2020; Laskin et al., 2014; Wang et al., 2019). FT-ICR-MS has been widely used in water-soluble organic carbon (WSOC) research using negative ion (–)ESI mode. The (–)ESI mode has a good response on components in WSOC with functional groups of –OH and –COOH (Li et al., 2022; Zhang et al., 2021; He et al., 2022). Until recently, HWM hydrophobic organic species, such as ester (ROR), hydrocarbons ($C_xH_y$), fat, and reduced Nitrogen-containing compounds, were limitedly understood in ambient PM samples. CHN and CHON compounds are favorably detected in positive ion ESI ((+)ESI) mode compared to (–)ESI mode (Lin et al., 2012). This study will investigate the processing of HMW carbonyls, esters, amines, and other nitrogen-containing compounds since they can form $[M+H]^+$ ions in electrospray ionization operation positive ion mode (Kanawati et al., 2008; He et al., 2021a).

We present the molecular description of the diurnal evolution of HMWOCs that is sensitive to the (+) ESI mode in FT-ICRMS analysis to explore a wider context of HMWOC. In this study, ambient $PM_{2.5}$ samples were collected in the Morning, Afternoon, and Night, as well as midnight and early Morning (MEM) in an urban area in Eastern China during spring for OA aging analysis. As a typical metropolitan area in China, the sampling location is influenced by

the local coal burning, biomass burning, traffic, and residual emissions, as well as long-range transport. The molecular-level characterization of HMWOC was explored, and organic subgroups' chemical composition and evolution under real-world conditions were identified. This study can expand the understanding of HMWOC's evolution in daily cycles and be supportive of evaluating the impact of organic aerosols.

## 2 Materials and Methods

### 2.1 Sample collection

The sampling site was located on the rooftop of a commercial building (119.0734° E, 33.6047°N) with a height of 45 m above the ground. The site is in a typical urban environment in Huanan, Eastern China, with roads, parks, and residential areas nearby. Emissions from restaurants, biomass, and coal burning from villages influence the region.

A high-volume sampler (Thermo Inc., USA) was used for $PM_{2.5}$ sampling at a flow rate of 1.13 $m^3$ $min^{-1}$. $PM_{2.5}$ was collected using a quartz filter (Whatman Inc. USA). The filter was pre-baked using a Muffle furnace at 600 °C to diminish organic species. Sampling periods are in the midnight and early Morning (MEM) (23:00–4:30), Morning (5:00–10:30), Afternoon (11:00–16:30), and Night (17:00–22:30). total of 52 samples were collected to describe the pattern of $PM_{2.5}$ chemical composition, and samples from April 20$^{th}$ 2021 to April 21$^{st}$, 2021 were selected for FT-ICR MS analysis.

The routine $PM_{2.5}$ chemical composition was analyzed, including carbonaceous species, water-soluble ions ($SO_4^-$, $NO_3^-$, $NH_4^+$, $Cl^-$, and $Na^+$), and elemental species. The protocols of these analyses are available in the literature and *supportive information* (Wang et al., 2018). The samples were punched from the quartz filter with an area of 0.526 $cm^2$. Then, the punches from

the same collection time were immersed using 10 ml acetonitrile with supersonic for 10 min three times. The acquired ACN solutions from the same sampling time were gathered, combined, and concentrated to 1 ml for FT-ICR analysis.

### 2.2 FT-ICR MS analysis

The MS analysis of organic aerosol was carried out on a 7.0 T Bruker Solarix 2XR FT-ICR mass spectrometer. The samples at a concentration of about 50 mg/L were dissolved in methanol for the ESI analysis. The samples were injected into the ionization source at 120 μL/h through a syringe pump. The typical operating conditions for positive-ion ESI analysis were: capillary voltage of –4.0 kV, capillary exit voltage of 200 V. Ions were accumulated in the hexapole for 0.05 s then were transferred into the ICR cell with a time-of-flight (ToF) of 0.7 ms. The ion transformation parameter for the quadrupole (Q1) was optimized at *m/z* 200. The mass range was *m/z* 150–1000. A total of 128 scans with 4M data points were accumulated to enhance the signal-to-noise ratio. Details of FT-ICR MS calibration and data processing are provided in *Supportive information*. Briefly, FT-ICR MS was calibrated using a reference list formed by the manually assigned known formula ($^{12}C_{0–100}$, $^{1}H_{0–200}$,$^{14}N_{0–10}$,$^{16}O_{0–20}$, and $^{32}S_{0–2}$.) in Data Analysis.

### 2.3 FT-ICR MS data-related parameters integration

A modified aromatic index ($AI_{mod}$) and double bond equivalent (DBE) were calculated for each assigned formula, according to Koch and Dittmar. (2006) The intensity-weighted average of elements (C, H, O, N, S), formulae (CHO, CHON, CHOS, CHONS), and other parameters (H/C, O/C, DBE, KMD, and $AI_{mod}$) were calculated for each sample. Molecular formulae were further assigned to the following groups as described by Seidel et al.(Seidel et al., 2014) and Antony et al(2015).

The $OS_C$ is used to describe the composition of a complex mixture of organics undergoing oxidation processes. $OS_C$ is calculated for assignable molecular formulae as follows (Kroll et al., 2011):

$$OS_C = -\sum_i OS_i \frac{n_i}{n_C}$$

Where $OS_i$ is the oxidation state associated with the element *I* and $n_i/n_C$ is the molar ratio of element *I* to carbon within the molecule. Other details of the data analysis are also available in *Supportive Information*.

## 3 Results and disscussion

### 3.1 Overview of molecular characterization of organic aerosols

The overview of the sampling period is available in *Supportive Information*. The concentration-time profile is shown in Figure S1. Although the mass concentration of $PM_{2.5}$ showed a fluctuating trend in a range of 32–81 μg m$^{-3}$, the fraction of organic carbon was stable with a range of 0.18–0.21(Figure S1). During the sampling site, wind came from almost all directions, epically from north, south, east, and partially west at the daily level (Figures S2 and S3). Moreover, the meteorological parameters were also stable during the sampling. Therefore, we picked one of these days to evaluate the daily evolution of HMWOCs as the representative sample.

During the observation, the average concentrations of organic carbon (OC) were 12.8 μg m$^{-3}$, following an order of 12.7 μg m$^{-3}$ (Night), 12.6 μg m$^{-3}$ (Afternoon), 11.1 μg m$^{-3}$ (Morning), and 9.4 μg m$^{-3}$ (MEM). The concentrations of secondary organic carbon were also estimated as

5.11μg/m$^3$ (46.01%), 4.97 μg/m$^3$ (39.32%), 5.59 μg/m$^3$ (43.8%), and 4.85 μg/m$^3$ (51.56%), following order of Night≈ Morning> Afternoon >MEM.

A summary of acquired parameters for four samples from FT-ICR MS is shown in Table 1, and the samples are marked with MEM, Morning, Afternoon, and Night. Among four samples, the number of formulas between 5781 and 6566 was detected, with an order of Morning<MEM<Afternoon<Night. The average molecular weights of all four samples were: Morning (393 Da)< Afternoon (395 Da)<MEM (402 Da)< Night (406 Da). The weighted element ratios, such as $O/C_w$ and $H/C_w$, were highest at Night, with values of 0.13 and 0.80, respectively, suggesting higher oxidation and saturation levels at Night compared to other three stages. The weighted carbon-normalized DBE ($DBE/C_w$) was the lowest at Night (0.21), also suggesting a high saturation level of organic compounds, possibly due to primary emissions of OA.

The reconstructed (+) ESI FT-ICRMS mass spectra of four samples are shown in Figure 1. Most organic compounds were found M.W. between 200 and 400 Da. All the compounds in the organic aerosol are clustered into oxygenates (CHO) compounds, oxygen- and nitrogen-containing (CHON) compounds, and reduced nitrogen-containing (CHN) groups. No sulfur-containing compounds were detected in the(+) ESI mode. On an intensity basis, the CHON is the largest subgroup, followed by CHO and CHN subgroups. CHON group accounted for 73% of the relative intensity in the MEM, and increased to 77%, 81%, and 93% in the Morning, daytime, and Night. CHN decreased from 10% to 3% after sunset. The result implies that daytime photooxidation significantly affected the chemical nature of HWOCs. In the following results and discussion, a detailed analysis would perform for the nature of daily HMWOC aging.

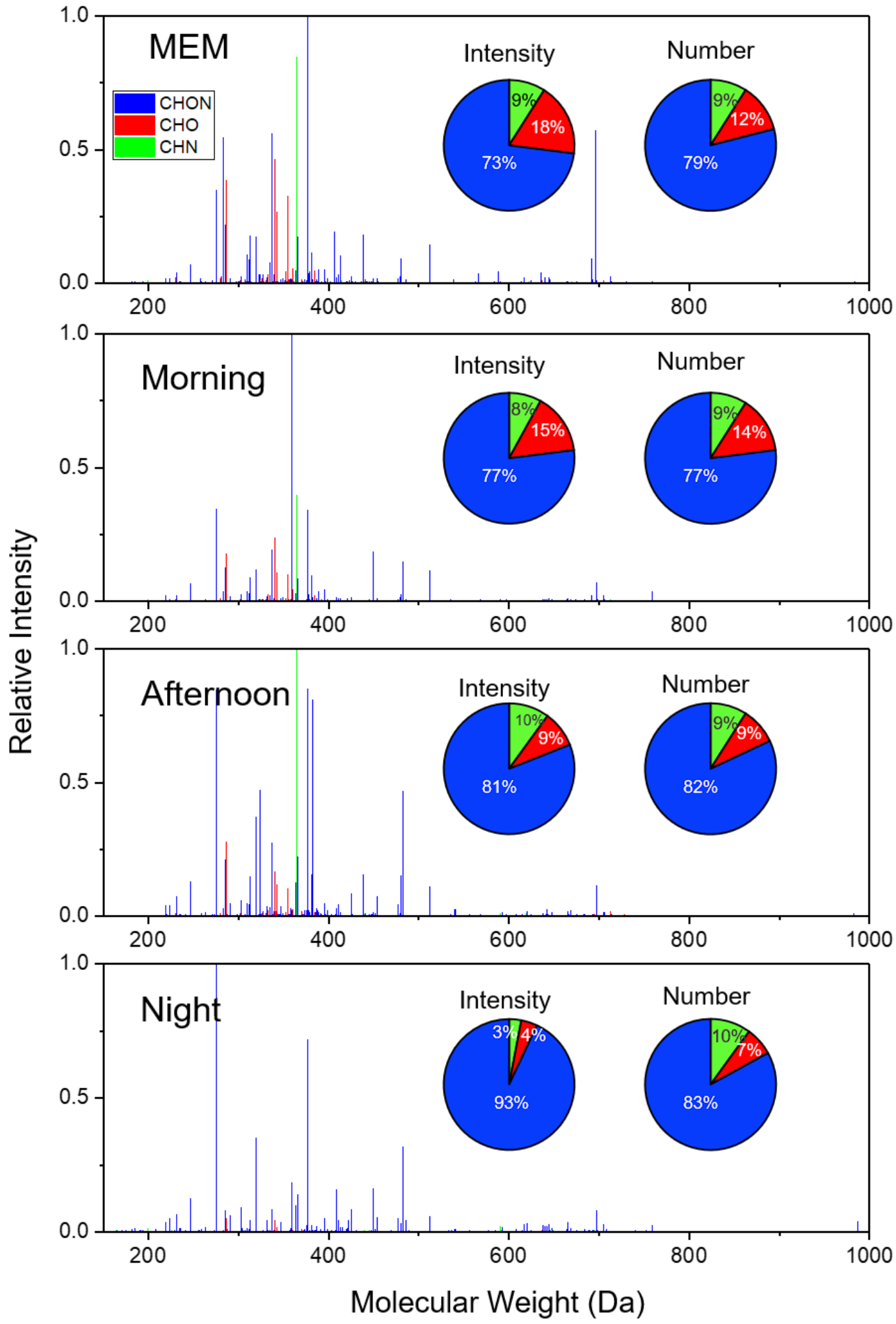

**Figure 1. Mass spectra and distribution of relative intensity and formula number of CHON, CHO, and CHN compounds detected by positive-ion ESI FT-ICR MS.**

**Table 1. Summary of molecular parameters of FT-ICRMS results among four samples during the different periods.**

| | | Number Frequency | Molecular Weight (Da) | $O/C_w$ | $H/C_w$ | DBE | DBE/C |
|---|---|---|---|---|---|---|---|
| Midnight and Early Morning | All | 5859 | 402 | 0.12 | 1.69 | 6.01 | 0.26 |
| | CHO | 715 | 345 | 0.13 | 1.47 | 6.88 | 0.31 |
| | CHN | 502 | 385 | 0 | 1.3 | 11.69 | 0.53 |
| | CHON | 4642 | 418 | 0.14 | 1.79 | 5.08 | 0.21 |
| Morning | All | 5781 | 393 | 0.12 | 1.66 | 6.35 | 0.26 |
| | CHO | 785 | 348 | 0.13 | 1.48 | 6.92 | 0.31 |
| | CHN | 531 | 482 | 0 | 1.43 | 12.23 | 0.41 |
| | CHON | 4465 | 402 | 0.13 | 1.74 | 5.66 | 0.23 |
| Afternoon | All | 6376 | 395 | 0.12 | 1.75 | 5.74 | 0.25 |
| | CHO | 606 | 368 | 0.13 | 1.47 | 7.03 | 0.31 |
| | CHN | 565 | 385 | 0 | 1.29 | 11.65 | 0.53 |
| | CHON | 5205 | 399 | 0.13 | 1.84 | 4.84 | 0.21 |
| Night | All | 6566 | 406 | 0.13 | 1.8 | 5.13 | 0.21 |
| | CHO | 432 | 429 | 0.12 | 1.46 | 8.22 | 0.31 |
| | CHN | 648 | 416 | 0 | 1.51 | 9.12 | 0.37 |
| | CHON | 5486 | 405 | 0.14 | 1.82 | 4.88 | 0.2 |

### 3.2 Evolution of CHO compounds and functionality

In the (+) ESI mode, the CHO group is favorably detected as carbonyl and ester compounds (Ditto et al., 2021). In number frequency, CHO increased from 12% (MEM) to 14% (Morning), then decreased to 7% until the Night. CHO molecules contained up to ten oxygen atoms, but most of the CHO compounds with an oxygen atom number ≤ 4, such as 74.7% in MEM, 73.5% in Morning, 69.5% in Afternoon, and 72.2% in Night.

As shown in Figure 2, a classification was performed based on the iteration of *Antony et al.* (2014). Saturated hydrocarbons, unsaturated hydrocarbons, lipids, lignins, polyphenols, and carbohydrates were detected in CHO. The saturated hydrocarbons, including lipids, alkanes, and aliphatics, were dominated, and they were mainly from traffic (Zhang et al., 2011), biogenic emissions, or biomass burning (Chen et al., 2014), alcohol synthesis(Ao et al., 2018), and biofuels (Rice et al., 2019). In a number frequency view, unsaturated hydrocarbons were 1.2 times more in MEM sample than in Morning, 1.3 times more than in Afternoon, and ~ 1 time in Night. Both samples' frequency was promoted by traffic emissions but removed by daytime photochemical activities (Mcguire et al., 2014).

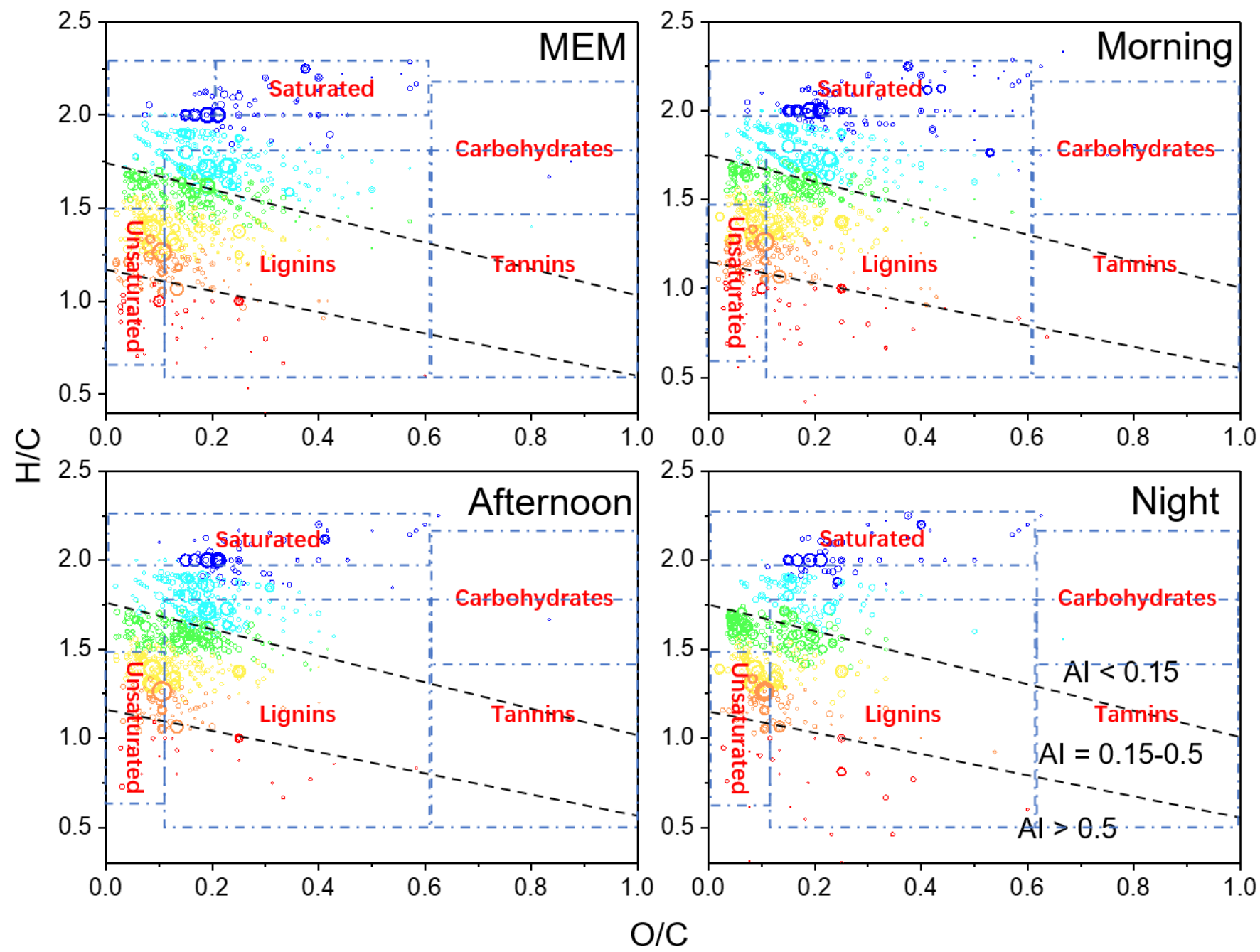

**Figure 2. Van Krevelen diagrams (H/C vs. O/C ratio) for CHO species with various aromatic index (AI) value ranges. The dashed lines separate the different AI regions. The size of the symbols reflects the relative peak intensities of molecular formulae on a logarithmic scale.**

Figure 3 illustrates the DBE against carbon number with color bars denoting the numbers of oxygen. The CHO compounds showed the highest DBE up to 26. Among them, CHO compounds with oxygen atoms ≤ 4, carbon atoms ≤ 25, and DBE between 4 and 16 were favorable phenol compounds with one or two aromatic rings (Yu et al., 2016). However, the HMW CHOs with carbon atoms ≥ 25

were commonly with oxygen atoms ≥ 6. They were recognized as HUMIC-like substances that were abundant with –OH, –COOH, and –CHO but an absence of nitrogen (Kurek et al., 2020). Figure S4 implies that the highest carbon oxidation state (OSc) occurred in compounds with a number of carbon atoms around 20.

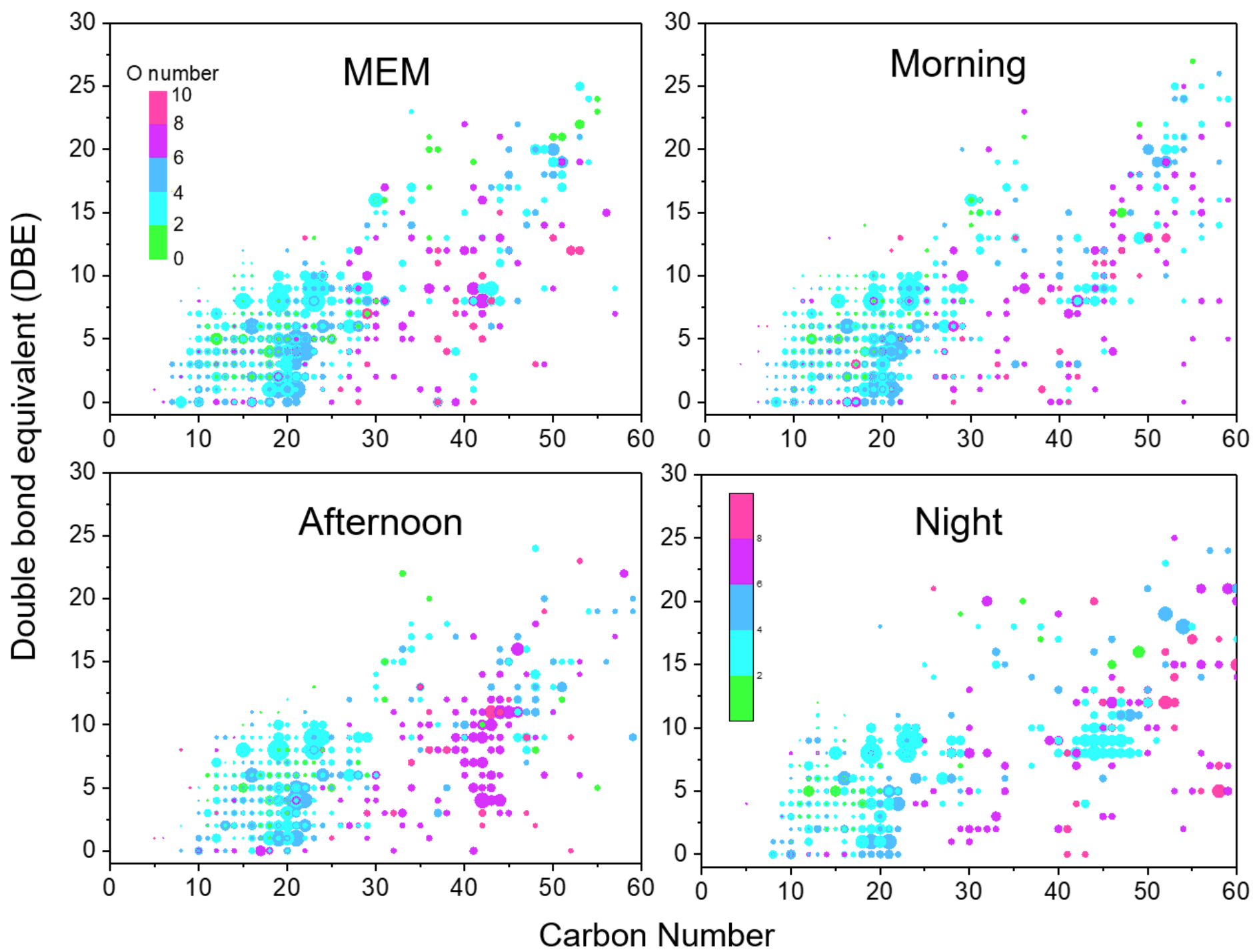

**Figure 3. Carbon number vs. double bond equivalent (DBE) for CHO species. The color bar denotes the number of O atoms. The size of the symbols reflects the relative peak intensities of molecular formulae on a logarithmic scale.**

The daytime photochemical activities extremely change the functionality of CHO compounds. For example, CHO with 20 carbon atoms was observed with more oxygen atoms in Morning and Afternoon samples than that in MEM and Night. Particularly, CHO compounds with carbon numbers between 40 and 50 were prominent in Afternoon, possibly due to the formation of –C=O and –COO via daytime photochemical reactions. Then, at Night, the highly oxygenated compounds (O number >6, DBE>5 and carbon atoms >50) appeared in the nighttime. Meanwhile, the batch of primary OA was also observed, such as CHO compounds with carbon atoms between 45–50, DBE between 5–10, and O atoms between 0 and 2. In addition, after serious aging, the highly oxidized CHO compounds with oxygen atoms larger than eight significantly increased in Night and MEM samples (Huang et al., 2014).

As shown in Figure 4, the KMD analysis of CHO compounds was performed. Many homologues of CHO compounds are present as the "core" molecules plus $(CO)_n$ (n= 0,1, 2, 3, …) as a result of carbonyl formation. Among these CHO homologs with $KMD_{CO}$ between 0.4 and 0.0, and oxygen atoms less or equal to four, carbonyl was favorably resistant or produced in the Morning vs. MEM scenario. However, in the Afternoon vs. Morning scenario (Figure 4b), as well as Night vs. Afternoon scenario (Figure 4c), the removal of carbonyls exited among CHO with oxygen atoms from one to ten. Likewise, the formation of carboxylic acids (–COOH), represented by $(COO)_n$ homologs, were significantly formed in Morning and removed in both Afternoon and Night, suggesting continuing destruction of carboxylic acids. Interestingly, the CHO compounds with oxygen numbers less or equal to two continued forming without solar radiation at Night. The removal of carbonyls could be attributed to further oxidation to carboxylic acids as well as reactions with $H_2SO_4$ and $HNO_3$ to create

organosulfate and organonitrates, or reply with ammonia to form reduced nitrogen-containing compounds (Liu et al., 2015) (Ervens et al., 2011).

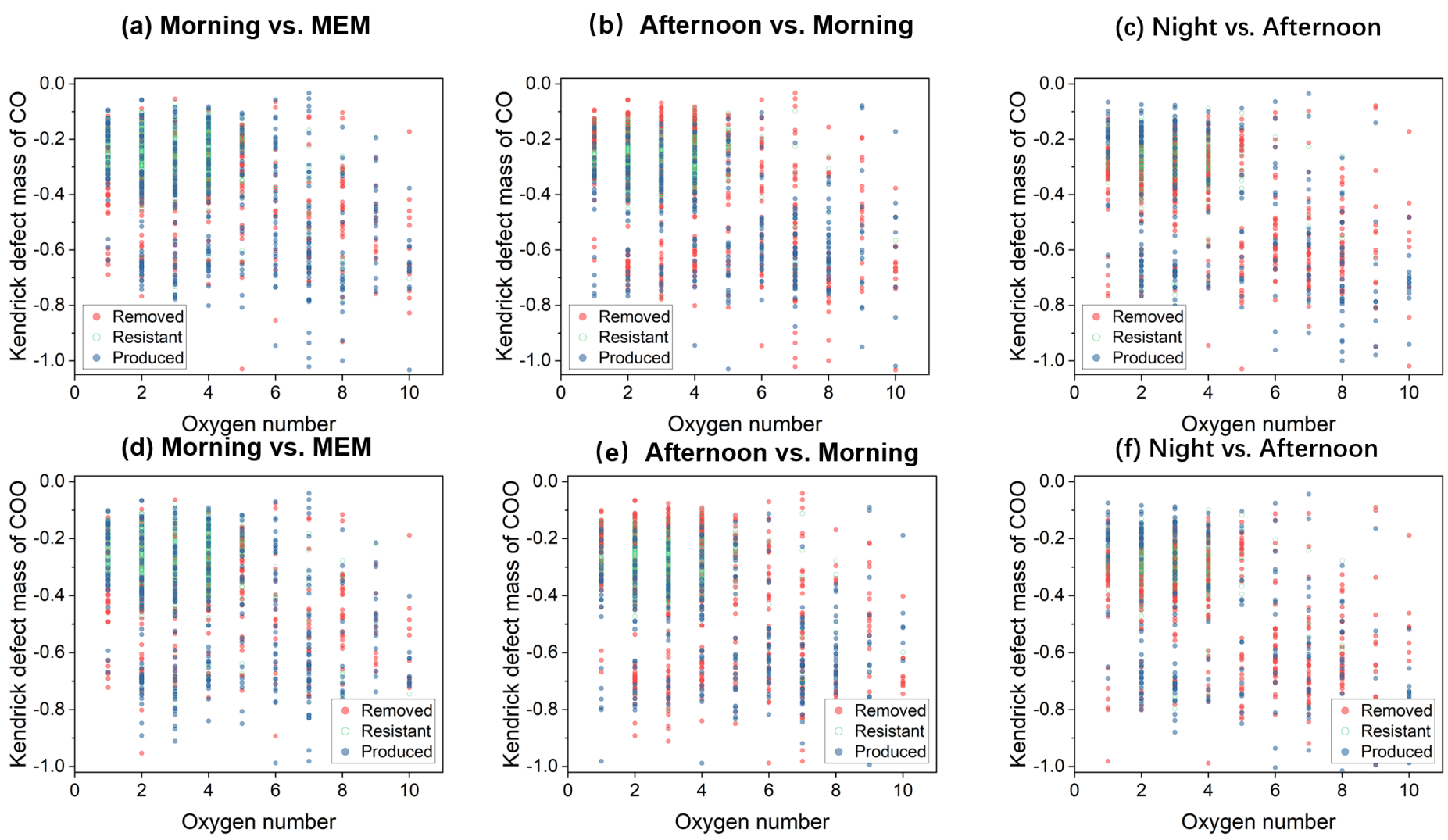

**Figure 4. Kendrick Mass Defect (KMD) plots of CHO compounds in diurnal evolution of CO series (a, b, and c) and COO series (d, e, and f)**

### 3.3 Evolution of CHON compounds and functionality

CHON compounds were the most abundant group in the (+)ESI results, accounting for 79%–83% in number frequency. The average molecular weight decreased during the daytime, from 418 da in MEM to 402 in Morning, and 399 in Afternoon, then increased to 405 at Night. $O/C_w$ varied from 0.13 to 0.14, while $H/C_w$ was between 1.74 to 1.82. $O/C_w$ was lower than 0.18, and $H/C_w$ was higher than 1.5. Those values were recommended for biomass-burning OA in (+) ESI mode (Song et al., 2022).

As shown in Figure 5, most CHON compounds, accounting for 60%–63%, were with $H:C_w > 1.7$ and AI< 0.15. These CHONs were mostly nitrogen-containing ones with $-NO_2$ or $-ONO_2$, namely unsaturated organonitrates. The unsaturated organonitrates were more pronounced in both MEM and Afternoon, taking up to 63.2% and 63.3%, respectively. The highly unsaturated (H:C=0.7–1.5 and 0.15<AI<0.5) decreased from 33.8% in Morning to 21.6% in Afternoon, possibly due to the influence of photobleaching from daytime photochemical activities(He et al., 2021b). Also, the aromatic CHON (AI>0.5) increased to 3.1% at Night.

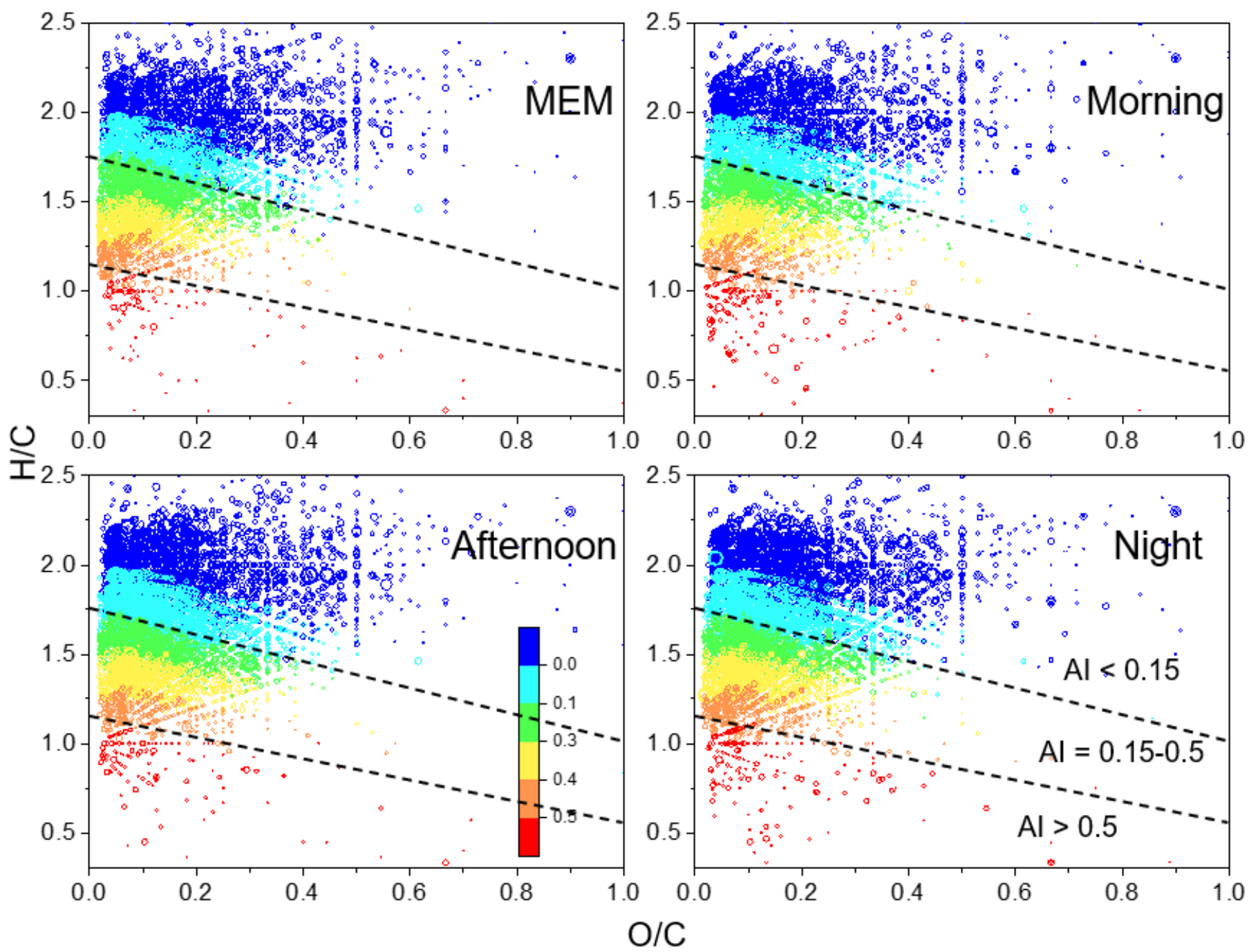

**Figure 5. Van Krevelen diagrams (H/C vs. O/C ratio) for CHON species with various aromatic index (AI) value ranges. The dashed lines separate the different AI regions.**

The OSc diagram against carbon number is shown in Figure 6. Most CHON compounds are distributed in a range of carbon numbers between 10 and 40 and OSc between –2 and 1. Those long-chain nitro-hydrocarbon with carbon numbers between 30 and 60 were substantially enhanced by the morning rush hours. During the daytime, CHON compounds with carbon numbers between ten and 30 and OSc between –1 and 0 were enhanced, suggesting that more aged nitro-hydrocarbons were produced in the daytime.

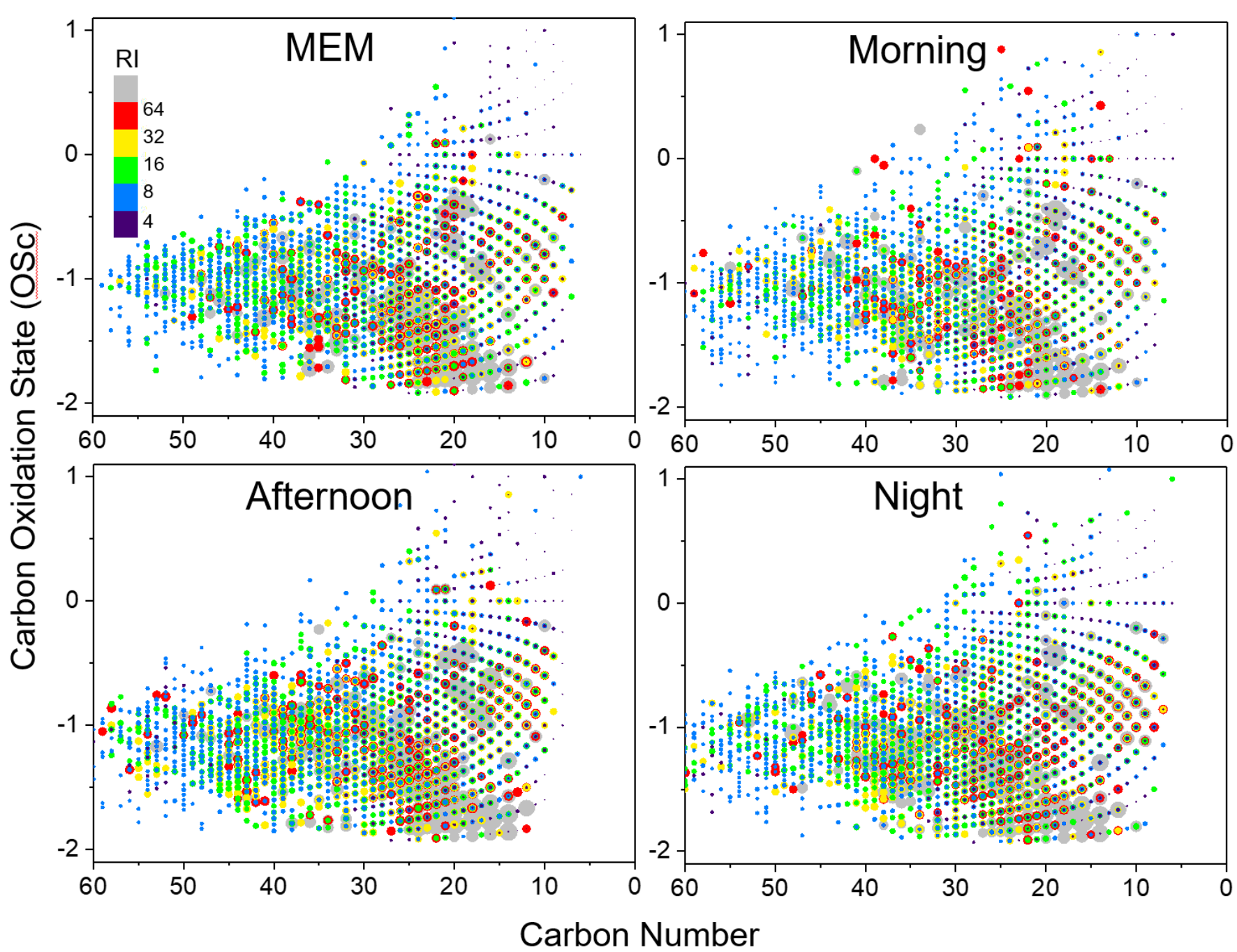

**Figure 6. Overlaid carbon oxidation state (OSc) symbols for CHON species. The size and color bar of the markers reflects the relative peak intensities of molecular formulae on a logarithmic scale. RI: relative intensity.**

Typically, CHON compounds with higher O/N ratios (≥3) can be attributed to organic nitrate ($RONO_2$) groups in +ESI mode (Song et al., 2022). The organonitrates accounted for 21.9%−24.0% of CHON compounds, and the associated ratios of log RI were between 22.2%−23.0%. $RONO_2$ compounds were more prominent in the Morning (24.0%), and lower without sunlight (e.g., 21.9% in MEM). $RONO_2$ compounds with AI>0.5, attributed to polycyclic phenolic compounds, escalated in the Morning (from 1.7% to 2.1 % after the rush hour), then decreased to 1.9% in Afternoon, and then increased to 2.0% at Night.

As shown in Figure S5, $N_1O_x$ (x=1–9) with DBE< 5 were the most abundant in the CHON subgroup, accounting for 49%–51% of CHON compounds in number frequency. Since most CHON compounds contained only one nitrogen atom, the $CHON_1$ group was chosen for further evaluation. Figure 7 shows that the $CHON_1$ $CH_2$-homologues with molecular mass between 150 and 400 were stable as being resistant, as shown in Figure 7 a–c. However, $CHON_1$ compounds with M.W. smaller than 200 were removed in Afternoon but produced in Night. The nighttime chemistry strongly removed the $CHON_1$ $CH_2$-homologues with m.w.> 400. These results suggest the diversities of atmospheric fate of $CHON_1$ compounds.

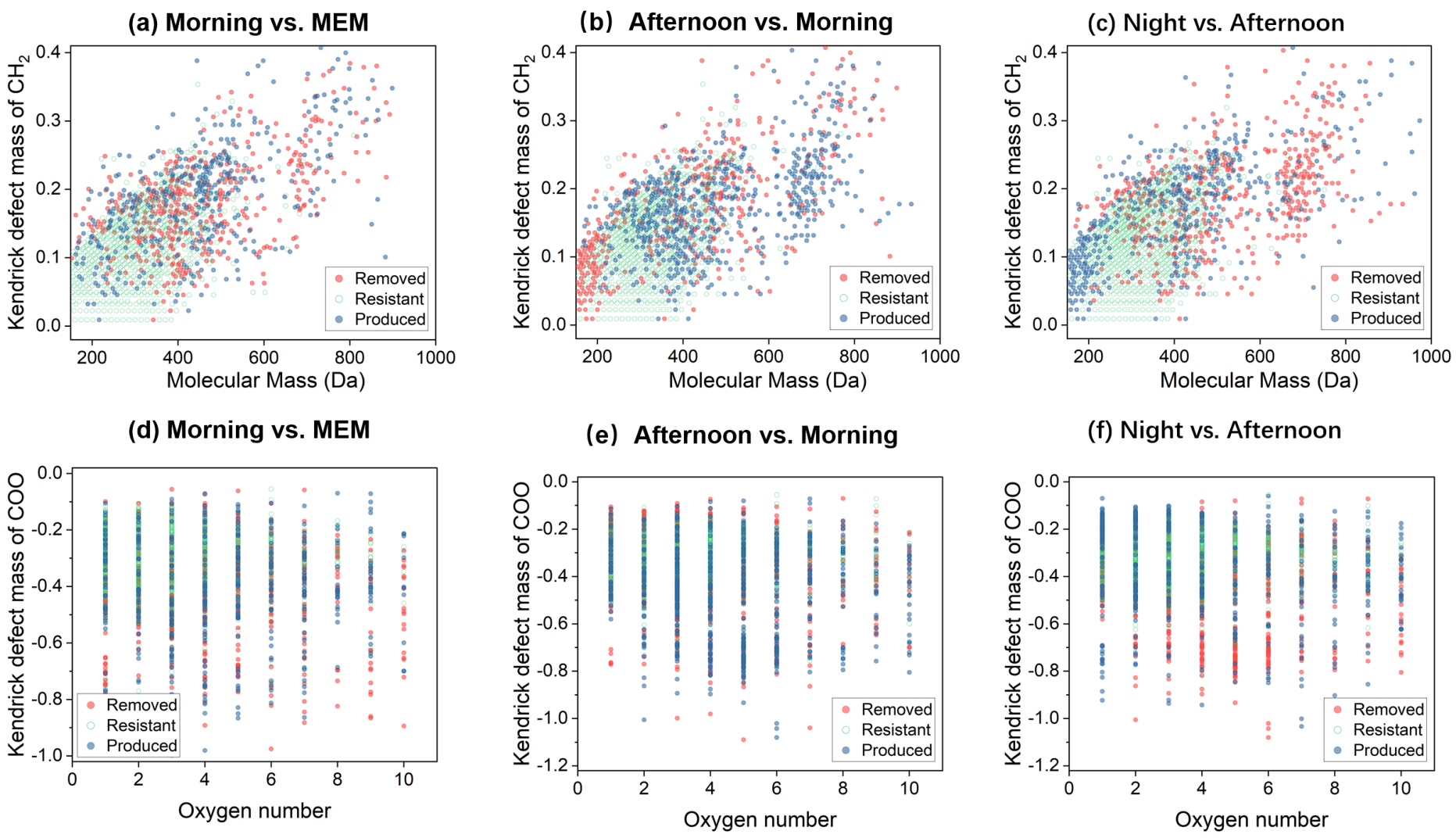

**Figure 7. Kendrick mass defect (KMD) plots of $CHON_1$ compounds in diurnal evolution of $CH_2$ series (a, b, and c) and COO series (d,e, and f)**

In an KMD view, as shown in Figure 7 (d–f), $CHON_1$ compounds (COO homologous) were removed in Morning, while produced prominently in Afternoon and Night (Figures 7(d) and 7(e)). Those results suggested that the organonitrates continued performing multi-generation oxidation to produce more carboxylic functional group (−COOH), and the carboxylic functional group can be removed due to the reaction with acids such as $HNO_3$ and $H_2SO_4$, or be neutralized by $NH_3$ in the particle phase.

### 3.4 CHN compounds' evolution and functionality

CHN compounds remained stable in the FT-ICR dataset, accounting for 9%–10% in number and 3%–10% in intensity. The average $H/C_w$ was between 1.30 and 1.51 among four samples. As shown in the DBE against carbon number diagram (Figure 8), most CHN compounds have carbon numbers between 10 and 60, as well as nitrogen numbers from 1 to 10. Only a trivial proportion (3.0%–4.5%) of CHN compounds with DBE=0 and carbon numbers>10 as long-chain aliphatic amines (Song et al., 2022). Long-chain aliphatic amines could be emitted from traffic and biomass emissions in the Morning, then decrease to 3.0% in the Afternoon, and finally reach 4.5% at Night due to rising relative humidity (Chen et al., 2019).

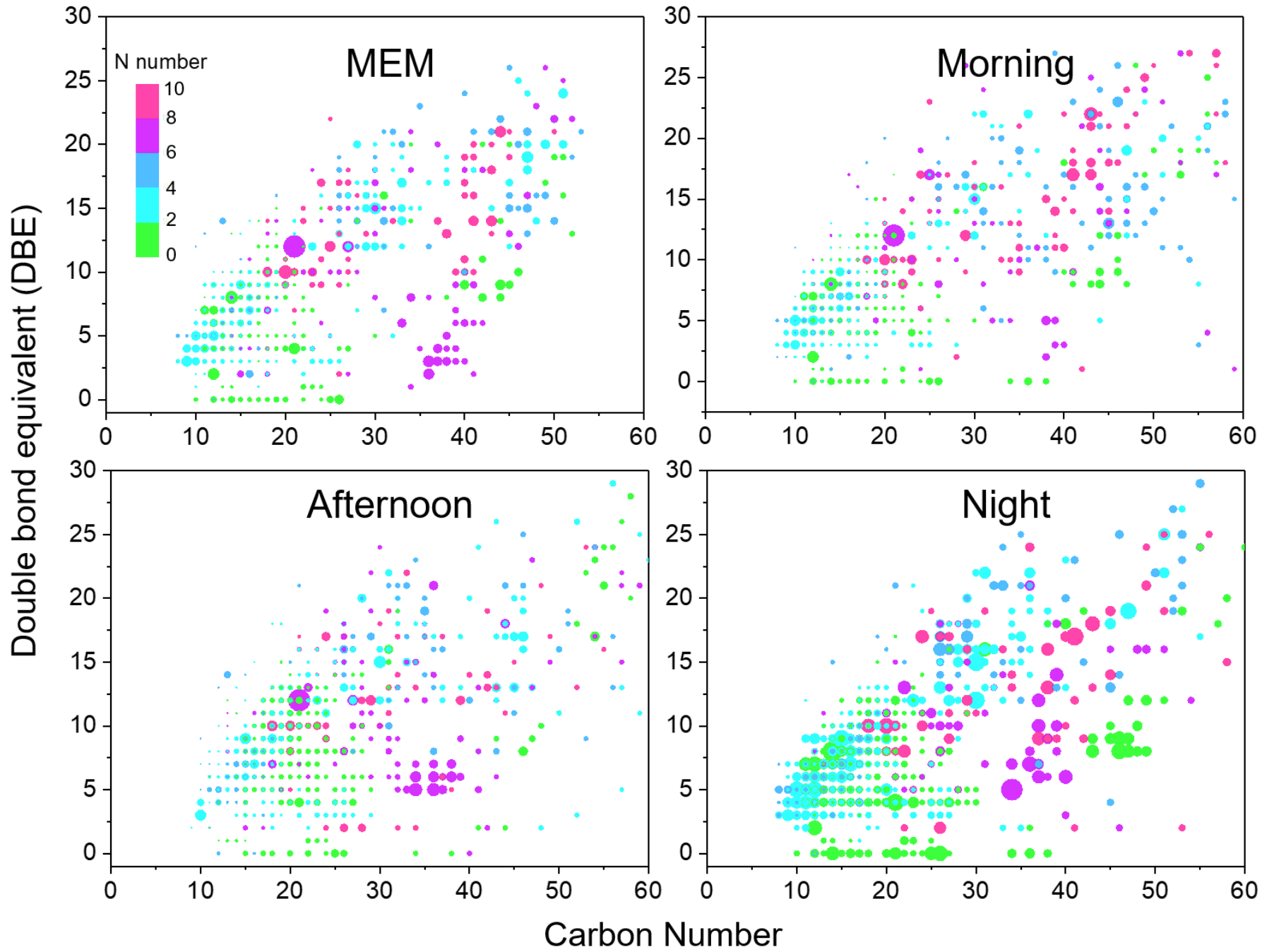

**Figure 8. Carbon number vs. double bond equivalent (DBE) for CHN species. The color bar denotes the number of N atoms. The size of the symbols reflects the relative peak intensities of molecular formulae on a logarithmic scale.**

CHN compounds with two nitrogen atoms ($N_2$) and DBE>5 were the most abundant group. For example, the series of $C_5H_8N_2(CH_2)_n$, $C_5H_6N_2(CH_2)_n$, and $C_7H_6N_2(CH_2)_n$ homologous series were likely imidazole, pyrazine/pyrimidine, and azaindole homologous series, respectively (Wang et al., 2019). 16%–18% of $N_2$-containing CHN compounds attributed to five-membered rings such as pyrazole, imidazole, and their derivatives, or six-membered rings N-heterocyclic species . The abundance of $N_2$-containing CHN compounds showed an increasing trend from 16.3% (MEM) to 27.2% at Night.

The nitrogen number($N_n$)>8 CHN compounds occurred in both day- and nighttime, suggesting the impact of the secondary formation of CHN compounds. After diurnal processing, the relative intensities of CHN compounds, with C numbers between 20 and 50, were enhanced. This result was consistent with the KMD analysis, as shown in Figure 9.

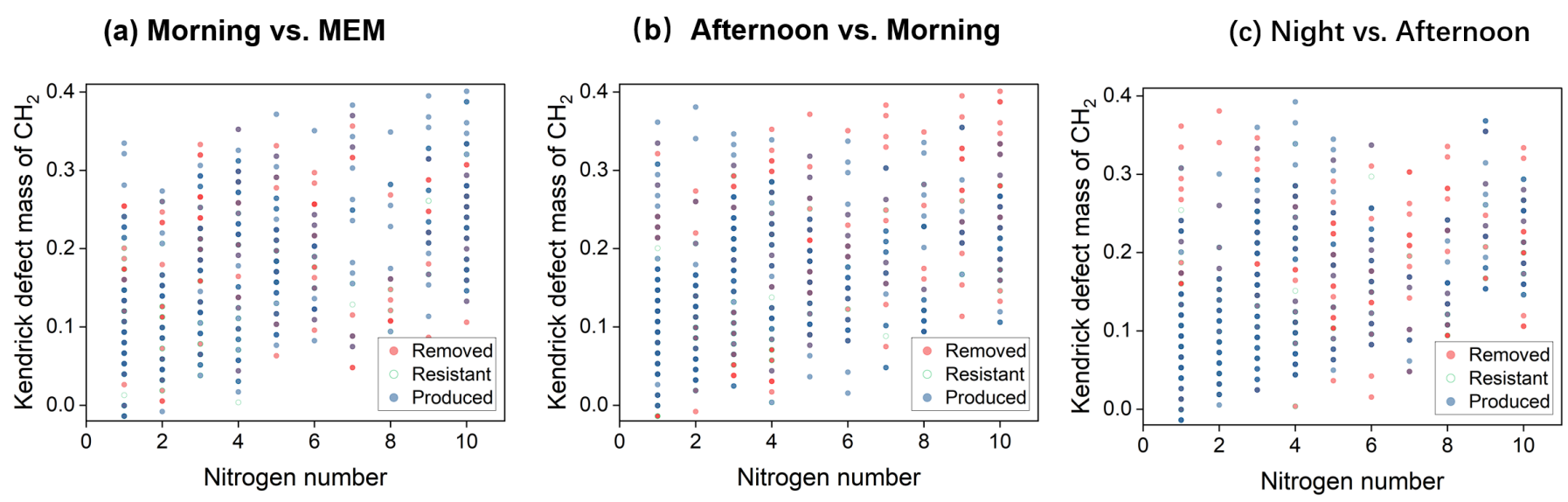

**Figure 9. Kendrick mass defect (KMD) plots of CHN compounds in diurnal evolution of CH2 series (a, b, and c).**

The $N_{8-10}$ containing CHN compounds accounted for 11.4%–18.5% among the four stages, appearing with homologous series. They typically appeared with carbon numbers larger than 20, with the largest abundance in MEM (18.5%). These CHN compounds can be vaporized from coal burning, especially in the smoldering stages. Therefore, the $N_{8-10}$ containing CHN compounds were reasonably more pronounced at low temperatures from residential coal use for heating. Besides the primary emission, small CHN compounds can also form from the reactions between amines/ammonia and carbonyl-containing SOA (Zarzana et al., 2012; Liu et al., 2015). Our results suggested that the formation of HMW CHN was also possible.

## 4 Atmospheric implications

We present the molecular description of the diurnal evolution of HMWOCs that is sensitive to the (+) ESI mode in FT-ICRMS analysis to explore a wider context of HMWOC.The uptake of ammonia on biogenic or anthropogenic SOA to form light-absorbing organonitrate has been reported (Li et al., 2017b; Liu et al., 2015). In this study, we proved that the nucleophilic oxygenated and nitrogen-containing compounds can also undergo substantial aging. Commonly, the nucleophilic HMWOC were considered less polar, and their evolution were different. We found more –CHO and –COOH groups were detected in the daytime for the CHO subgroup, and the aerosol-phase oxidation continued in the dark, resulting in more oxidized HMWOCs.

The formation of HMW organonitrates was also complicated. We have observed the prominence of organonitrates in both day and Night. In the daytime, the possible pathway includes the reactions of $NO_2$ radicals with aliphatic hydrocarbons and aromatic rings. At nighttime, the adduct of $NO_3$ radical on HMWOC could possibly occur. In polluted urban areas, the rich nitrogen oxides ($NO_x$) environment produced nitrate as well as organonitrates. The processing can shift the nitrogen deposition and be important on a local or regional scale. Also, the $CHON_1$ compounds were along with carboxylation, suggesting that the multiple oxidations of organonitrates can occur in both day- and nighttime. The process can enhance hygroscopic growth and CCN activities due to the acid-base reactions between ammonia and organics with carbonyls(Dinar et al., 2008).

The daytime CHN compounds could produce from the condensation reaction between –CHO groups and ammonium/amines. It might propose that the formation of long-chain CHN might be less

important, however, those species have possibly five- or six-member rings that can act as chromophores. Along with the primary emission of HMW CHN from coal combustion and biomass burning (Brege et al., 2018; Ray et al., 2019), the sources of BrC should be reconsidered. Moreover, the impact of ammonia was commonly on its potential impact on nitrogen deposition or forming the secondary in organic aerosol, but we suggest assessing its effect on the formation of HMW BrC. Conclusively, the evaluation of nucleophilic HMOC properties due to variability should be considered in future studies for improvement of air quality and climate model performances.

**Data Availability**

Data supporting this paper can be found at https://doi.org/10.5281/zenodo.7830299.

**Author Contribution**

YC designed this study. QL, CH, MH, and HChe contributed to the data collected during the field campaign. JZ, SZ, and YX performed field experiments. YL, XY, YH, XL, LM, QS, CC, FY and YC contributed to the scientific discussion and paper correction.

**Competing interests**

The contact author has declared that none of the authors has any competing interests.

**Acknowledgments**

The authors are grateful for the assistance of colleagues for sample collection.

**Financial support**

This research has been supported by the National Key Research and Development Program of China (grant no. 2018YFC0214001) and the National Natural Science Foundation of China (grant no. 42075109).

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
