# Peer review of "Evolution of Nucleophilic High-molecular-weight Organic Compounds in Ambient Aerosols: a case study"

_EGUsphere, 2023_

## Author Comment (AC1)

Dear reviewer,

We appreciate your comments and time on this manuscript, and we found the comments very helpful for us to improve the presentation of our data and interpretation. Besides most of your concerns about the "more robust sample-to-sample comparisons and statistics," we prepared the following analysis in the text and SI.

Indeed, the complex processes need to be carefully considered. Figures in supportive information illustrate the characteristics of the overall chemical composition of $PM_{2.5}$, meteorological conditions, and near-surface wind speed and wind direction to provide a whole picture of the study. By following the requirement of the reviewer, we performed a comparison of formula numbers between all 52 samples, and we found that the diversity of results of FT-ICR MS was "stable and typical," as shown in Figure 1 in the reply. Based on the information, we are confident that the samples used in this manuscript can give a snapshot of the organic aerosol evolution in the region. Certainly, the analysis can cause uncertainties due to the limited amount of samples, and we discuss the limitations of the work in Atmospheric Implication to avoid possible misguidance. In addition, we have added the "case study" statement in the title.

For the specific comment, we have also prepared a point-to-point response to the comment, and the corresponding changes in the manuscript are also marked in the revision. We hope our efforts can meet your standard for publication. Again, all authors are thankful for the comments.

**Reviewer 1**

My major concern with this paper is that it seems like one day of samples (four in total collected over one day) was used as a representative set for the entire field campaign (described at lines 96 and 139). The paper then proceeds to compare morning, afternoon, night, and overnight time periods with respect to various molecular features, but without more than one sample per time period, it is impossible to know whether any of

the trends that are discussed are meaningful. With highly variable emissions, oxidation conditions, and influx of air parcels carrying a wide range of fresh and oxidized species from different sources/oxidation conditions themselves, more data is needed to make reasonable conclusions about atmospheric processing given the extremely complex processes that the manuscript is trying to describe. Unfortunately I think this paper should be rejected in its current form and re-submitted upon inclusion of several more samples. It looks like 52 samples were collected over several weeks, so I'd like to see some of those incorporated into this analysis for more robust sample-to-sample comparisons and statistics. This work has the potential to yield very interesting results about the evolution of high molecular weight organic compounds, and I hope with more data we can learn more about their atmospheric chemistry.

Introduction:

Line 38: do you mean "due to the condensation of semivolatile vapors onto primary OA"? This part of the sentence is unclear.

**Response:**

Thanks for the suggestion. We have revised the sentence into (line 37):

"The condensation of semivolatile vapors that are more oxidized onto primary OA can lead to the evolution of OA during their atmospheric lives (Jimenez et al., 2009), and the evolution can also be caused by heterogeneous processes in aerosol phase (Ervens et al., 2011)."

Line 43: Suggest defining HULIS prior to using the acronym.

**Response:**

We have apprehended the definition of HULIS

Line 43: "Humic-like substances (HULIS)."

Line 46: I would consider phrasing this as "can generate nitrogen- and oxygencontaining organics" for clarity.

**Response:**

We have changed it (lines 46-47): "…can generate nitrogen- and oxygen-containing organics"

Line 57: I would suggest adding a caveat here, e.g., "as a class of potentially light-absorbing…" since these properties depend heavily on chemical structure.

**Response:**

Accepted and changed (line 58).

Line 63: Suggest defining HUMIC before using the acronym.

**Response:**

Thank you, we have defined it in line 43.

Line 66: Suggest defining ESI before using acronym.

**Response:**

"ESI" has been defined as "electrospray ionization (line 68)".

Line 69: Do you mean "fatty acid"?

**Response:**

Thanks for the reminder. "Fat" is not a scientific term; we have revised it into fatty acid (lines 70-71).

Line 73: Perhaps could say "other nitrogen-containing functional groups" for clarity.

**Response:**

Revised as suggested (line 74).

Line 77-78: The sampling period labels are a bit confusing here but they make more sense in the Methods section when you describe which time of day they correspond to. Perhaps you can say "samples were collected for ## hours across the diel cycle" here

for clarity.

**Response:**

We have rephrased the sentence to:

"In this study, ambient $PM_{2.5}$ samples were collected for 5.5 hours for four samples in a daily cycle (lines 78-81)."

Materials and methods:

Line 95: Capital "T" missing in "total".

**Response:**

Corrected.

Line 96: Are you only considering 1 day's worth of samples for FT-ICR MS analysis? I strongly suggest using more than 1 day's worth of data here. A 52-sample dataset is strong; why weren't more days included here? Over how many days were those samples collected? This information is in Figure S1 but you should mention it in the main text.

**Response:**

Indeed, the inter-comparison between the results of different FT-ICR MS is consuming. We will emphasize that our study is mostly a case study and carefully laid out the limitations of this study in the Atmospheric Implication part.

By following the requirement of the reviewer, we performed a comparison between samples, and we found that the diversity of results of FT-ICR MS was "stable and typical," as shown in Figure 1 in this response.

[Figure]

Figure 1 Ratios of formula numbers among the sampling period

The information in SI was added in Section 3.1 (Line 143), and the Figure 1 was also included in SI.

Line 102: Should be "sonication" instead of supersonic. Also, how many punches did you combine together? "The punches from the same collection time were immersed" sounds like you combined punches for extraction. If you use "ACN" later in the paragraph, please define it here.

**Response:**

We have revised "supersonic" to "sonication" in the corresponding sentence. The definition of ACN (Acetonitrile) has been added to the sentence (line 105).

Line 107: Can you clarify whether your acetonitrile extracts were then mixed with methanol for ESI analysis? If so, how do you know the concentration since these are field samples?

**Response:**

The sentence has been rephrased to:

"The mixture of acetonitrile extract was vaporized to dry. Then, the dry extract was reconstituted with water, and methanol was added proportionally in the water solution before FT-ICR MS analysis (lines 109-111)."

Line 115: I would like to see more description of this calibration procedure. Is this done in post-processing?

**Response:**

We have apprehended the following sentences into the SI:

The mass spectrometer was initially calibrated using sodium formate and then recalibrated with a known mass series in natural organic matter (Suwannee River fulvic acids), which contains a relatively high abundance of CHO formulae, providing a mass accuracy of 0.2 ppm or higher throughout the mass range of interest (line 20 in SI).

Line 121: All acronyms need to be defined (KMD, AI are new).

**Response:**

Sorry that we have moved the definition into SI. We have added the full description of them in the revision.

"Kendrick mass defect (KMD) were calculated according to Stenson et al." has been added at lines 123. The definition of AI is also apprehended (line 125).

OSc part: did you include only CHO species here, or are N and S included too? N and S can have different possible oxidation states and are not straightforward to include here unless you know the functional group they're present in (which you might). If you included them, I'd like to see more discussion on your approach. If you did not include them, I'd like to see more discussion on the impacts of leaving them out, given how important they are to your overall distribution (CHN and CHON especially, not really S since you mentioned you did not see it).

**Response:**

When calculating OSc, the two elements, Nitrogen and Oxygen, were considered. In

positive ESI mode, the valence of N was considered as -3. Also, S element was not considered because no S-containing molecules were detected in the sample in positive ESI mode. Therefore, we didn't put S into OSc calculation in CHN and CHON groups.

We have added this part of the explanation to the revision (lines 136-138).

**Results and discussion:**

Line 136: Should be "sampling period" not "sampling site".

**Response:**

We have changed it (line 146).

Line 137: Typo, should be "especially".

**Response:**

Corrected (line 146).

Line 139: Emissions will change throughout your sampling period, as well as air parcel histories (as shown by your wind rose with air coming in from all directions). 1)Which meteorological conditions are you referring to and do you have data to show us that displays how stable they were during sampling? I do not think it is valid to select one day of samples and draw conclusions from that one day, given the immense expected variability in emissions locally, as well as air coming in from other locations, and possibly varying chemical conditions locally/regionally as well. Why was only one sample selected here? This weakens the paper and limits the conclusions you can make. If only one day's worth is available for analysis, I think the paper needs to be re-framed as a case study, with the limitations of having only one day very clearly laid out.

**Response:**

We thank the reviewer for this valuable question.

The representativity is the most important thing we have considered during the

preparation of this manuscript. As shown in Figure 2, wind direction and wind speed shifted in a typical diurnal cycle that varied in all directions. The windrode in Figure 1 is similar to Figure S1.

[Figure]

Figure 2. Windrose plot of the sampling day

Following the suggestion from the reviewer, we also performed an inter-comparison between different samples (Figure 1, provided above). We found that the proportion of CHO, CHON, and CHON remained stable, and the average of parameters of samples were close to the period statistics. Therefore, we are confident that the chosen day (20th April) can represent the chemical nature of the organic species during sampling periods.

We have added the following part in Atmospheric Implications:

"It is noticeable that there were some limits on this work. Under the variable emission, atmospheric process, and long-distance transport of organic aerosols, the samples can give a snapshot the evolution in the region, and the analysis can cause uncertainties.

Therefore, this work can be treated as a case study." (Section 4)

Line 141-145: If this is based on just one day, it is challenging to draw meaningful conclusions/trends from any of these data. Comments from here on out don't address the data trends themselves since it is difficult to interpret these trends without more samples.

**Response:**

We provided enough evidence to convince the reviewer.

Line 160: Could you comment on how ionization efficiency might be impacting these statements?

**Response:**

The ionization efficiency is related to the molecular structure. In positive ion ESI mode, basic nitrogen compounds such as ammonia or pyridine compounds are easily ionized. Besides, ketone and ester groups can also be ionized. At the same concentration, the higher the ionization efficiency, the easier the compounds are ionized. Among the above compounds, ammonia has the highest ionization efficiency, followed by pyridine and, finally, ketone or ester. However, the ionization efficiency is also affected by other molecular structures, such as the steric hindrance due to the position of the side chain. So, in this study, most of the compounds detected are CHN or CHNO compounds.

Line 184: I noticed that here you talk about number frequency instead of intensity. It would be good to be consistent throughout your paper or explain why you are discussing different frequencies at different times.

**Response:**

The two terms, number frequency and relative intensity of formula were used with different conditions. This study uses number frequency to describe the complexity and diversity of organics among $PM_{2.5}$ samples. The normalized intensity can be used to evaluate the relative changes between different $PM_{2.5}$ samples. Therefore, we used both terms in different part of the description and discussion.

Line 186: I'd like to see more description of the meteorological conditions during sampling so we can get a better idea for what kind of photochemical conditions existed.

**Response:**

The summary of meteorological conditions is shown in the following Table 1.

Table 1. Summary of meteorological conditions

| Temperature (°C) | Dew point (°C) | Pressure (hPa) | Wind direction (° ) | Wind speed (m/s) |
|---|---|---|---|---|
| 14.1 | 8.7 | 1018.1 | 70.4 | 2.2 |

It was in a typical spring photochemical condition.

Line 210: Are these necessarily POA of #O is 1-2? Could they be SOA after 1-2 generations of oxidation?

**Response:**

We agree with the reviewer that the OA of #O 1-2 can partially be SOA after oxidation. Our statement was mainly based on the local emission pattern that coal and traffic from heavy-duty diesel vehicles that was only allowed to inter the urban areas at Night.

Then we have changed the statement into:

"The batch of primary OA and less-oxidized SOA was also observed, such as CHO compounds with carbon atoms between 45–50, DBE between 5–10, and O atoms between 0 and 2." (lines 220-221)

Line 245: As mentioned above, did you account for N in these OSc calculations, as it can have different oxidation states depending on the functional group structure? Since you see so much CHON, I suggest discussing how you included it or why you excluded it and what impacts that might have on your results.

**Response:**

When calculating the OSc of CHON-like compounds, the oxidation state of N is also

taken into account, and N valence was -3.

In general, I found the figures interesting but containing a very large amount of data without much guidance to readers about which parts we should focus on. Many of the panels looked very similar at least without zooming in on interesting regions or circling parts that we should pay attention to. I suggest adding some information to help guide readers through these panels and to help point out differences between panels that we should focus on.

**Response:**

Thanks very much for the suggestion. Indeed, we have a very complex data matrix to interpret. In the revision, combined with the suggestions from reviewers, we also improved the presentation of data for readability.

SI:

Lines 25-29: Are there references for these constraints?

**Response:**

The following references have been added in the SI:

He, C., Zhang, Y., Li, Y., Zhuo, X., Li, Y., Zhang, C., Shi, Q., 2020. In-House Standard Method for Molecular Characterization of Dissolved Organic Matter by FT-ICR Mass Spectrometry. ACS Omega 5(20), 11730-11736.

Kujawinski, E.B., Behn, M.D., 2006. Automated analysis of electrospray ionization fourier transform ion cyclotron resonance mass spectra of natural organic matter. Anal. Chem. 78(13), 4363-4373.

Lines 30-36: Are there references for these compound class attributions?

**Response:**

The following references have been added in SI:

Seidel, M., Beck, M., Riedel, T., Waska, H., Suryaputra, I.G.N.A., Schnetger, B., Niggemann, J., Simon, M., Dittmar, T., 2014. Biogeochemistry of dissolved organic matter in an anoxic intertidal creek bank. Geochim. Cosmochim. Acta 140(1), 418-434.

Antony, R., Grannas, A.M., Willoughby, A.S., Sleighter, R.L., Thamban, M., Hatcher, P.G., 2014. Origin and sources of dissolved organic matter in snow on the East Antarctic ice sheet. Environ. Sci. Technol. 48(11), 6151-6159.

**References**

Seidel, M., Beck, M., Riedel, T., Waska, H., Suryaputra, I.G.N.A., Schnetger, B., Niggemann, J., Simon, M., Dittmar, T., 2014. Biogeochemistry of dissolved organic matter in an anoxic intertidal creek bank. Geochim. Cosmochim. Acta 140(1), 418-434.

Antony, R., Grannas, A.M., Willoughby, A.S., Sleighter, R.L., Thamban, M., Hatcher, P.G., 2014. Origin and sources of dissolved organic matter in snow on the East Antarctic ice sheet. Environ. Sci. Technol. 48(11), 6151-6159.

He, C., Zhang, Y., Li, Y., Zhuo, X., Li, Y., Zhang, C., Shi, Q., 2020. In-House Standard Method for Molecular Characterization of Dissolved Organic Matter by FT-ICR Mass Spectrometry. ACS Omega 5(20), 11730-11736.

Kujawinski, E.B., Behn, M.D., 2006. Automated analysis of electrospray ionization fourier transform ion cyclotron resonance mass spectra of natural organic matter. Anal. Chem. 78(13), 4363-4373.

---

## Author Comment (AC2)

Dear reviewer,

We highly appreciate your precious comment on our manuscript and your time. We found the comment can substantially improve the quality of our manuscript and make it possible for an appropriate scientific presentation. We believe that your comments, particularly on the MS data quality control, operation conditions, interpretation of the result, and the possible limitation of this study, were substantially valuable. We have prepared a point-to-point response to the comment. We hope our revision can satisfy both the reviewer and the *ACP* journal.

**Reviewer 2**

Review of "Evolution of Nucleophilic High-molecular-weight Organic Compounds in Ambient Aerosols" by He et al.

General comments: In this manuscript, the authors present a detailed characterization of the molecular composition of aerosol samples collected throughout the day in China. They use FT-ICR MS analysis to identify molecular formula for each sample and make intercomparisons between the samples. Overall, there is a good deal of data being analyzed and the samples appear interesting. However, there are substantial gaps in the explanation for the methods as well as some of the interpretations that must be addressed before this work is published. I consider these major revisions and these are listed in the specific comments below.

Specific comments:

More detail needs to be included on how the peak identification was carried out. In addition, a peak list that includes the measured m/z, the proposed molecular formula and the corresponding exact mass, and the mass error should be included in supplemental information for each sample. Other items that need to be explained include:

How were isotopes handled? Are they included in the peak identification or removed

initially? If they were removed, what criteria were used?

**Response:**

In this study, $^{13}C$, $^{18}O$ and $^{15}N$ isotopic molecules were assigned and analyzed when molecular formula assignment was performed.

Why was sodium not included as a possible component of the molecular formula? Sodium comes from the samples as well as the glassware that is used and it is a very common adduct to see in positive ion mode ESI. If you did not include sodium in your peak identification list I strongly encourage you to go back and re-identify the peaks with it included. It is possible that many of the higher number N containing ions are actually just ions with sodium as the charge carrier.

**Response:**

The $[M+Na]^+$ ion is present in the original mass spectra. The Na adducts ions were also considered in the molecular formula assignment. Considering that the number and relative abundance of $[M+Na]^+$ ions are lower than those of $[M+H]^+$ ions (Figure 3), only [M+H]+ was counted and analyzed.

[Figure]

Figure 3 Mass scale-expanded segments at *m/z* 296 of the (+) ESI FT-ICR mass spectra of sample MEM.

Why was the max O/C set at 1.5? This limits the peak identification of things like sugars and levoglucosan which can come from biomass burning. It is fine if these were not observed, but that limit should not be set at the beginning.

**Response:**

We set the maximum value of O/C at three before proceeding to molecular formula assignment. Only very few molecules with O/C > 1.5 were detected in samples #10 and #11 (please see the single Excel table), and the relative abundances of these molecules are so low that they do not affect the overall conclusions of this paper.

What percentage of the total peaks were identified and what percentage of the total signal?

**Response:**

We are sorry that we did not understand the question. The total peaks took up to, of course, 100%.

How were blanks collected and handled in this analysis? How were blank ions removed from the list of identified peaks?

**Response:**

There are two types of blanks that we handled in the analysis.

The first is the blank from a blank filter for $PM_{2.5}$ sampling. The filter belongs to the same batch of filters for $PM_{2.5}$ collection. This blank can indicate the possible organic residues on the filter. The second is the instrument blank. After the ion source of FT-ICR MS was sufficiently cleaned, the instrumental blank was acquainted with the sample of pure LC-MS methanol.

During the data analysis, we treated the blank with caution by not directly removing the peaks from the blank. Firstly, the sample is from airborne material, and some substances can co-exist in both the blank and samples.

Subtraction between samples and the blank using relative intensity was inappropriate neither due to the competitive nature of ionization between various components in ESI mode. The peaks from the blank might be higher than those in samples, leading to negative values after the subtraction. Therefore, we compared the peaks manually to

determine whether the substances in the samples were originally from the blank.

The paper terms these nucleophilic HMWOC but no justification is provided for the term nucleophilic. Why is this terminology being used here? Organic compounds in atmospheric aerosols are often multi-functional, so care should be used in naming because this study is only reporting molecular formula.

**Response:**

We agree with the reviewer on this issue. Yes, when we use the term "nucleophilic", we think the positive ESI mode can be more sensitive to the compounds that are nucleophilic, such as ketones, carbonyls, and reduced nitrogen-containing compounds. Indeed, those compounds were often multi-functional. The nucleophilic term has to be used with caution.

Carbon oxidation state is a method used to intercompare organic molecules with carbon, hydrogen, and oxygen atoms in the molecular formula. If you are calculating this for chemicals that contain nitrogen, you need to include estimates of the oxidation state of nitrogen. The analysis for the CHO compounds is fine, but the later analysis for the CHON compounds (Figure 6) needs to be corrected.

**Response:**

When calculating OSc, the two elements, Nitrogen and Oxygen, were considered. In positive ESI mode, the valence of N was considered as -3. Also, S element was not considered because no S-containing molecules were detected in the sample in positive ESI mode.

The OSc calculated in Figure 6 followed the principle mentioned above.

Figures S1-S3 show some summary information from the study but it is not clear if the data from S2 and S3 are for the shorter window used for FT-ICR analysis or for the whole range shown in Figure S1. Please clarify.

**Response:**

We are sorry for the confusion. Figures 1-3 were presented for the representative of the data. All three Figures showed the whole duration of the sampling period. We have changed the caption of the figures to clarify.

On page 7-8 the concentration of organic carbon is given with estimates of the secondary organic carbon. How was the secondary organic carbon estimated (what method was used)? Why is the certainty in the concentrations higher than the certainty for the organic carbon (more units past the decimal)? What do the percentages signify that are listed after each value?

**Response:**

The method we used to estimate the secondary carbon can be found in one of our previous work (Wang et al., 2018). Briefly,

The EC-tracer method has been widely used to estimate SOC (Turpin and Lim, 2001), which can be defined as

$$POC= (OC/EC)_{prim} \times EC \ (1)$$

$$SOC= OC–POC \ (2)$$

where POC, SOC, and OC stand for the estimated primary OC, secondary OC, and measured total OC, respectively.

We have put this part into the SI (lines 60-65).

And the digits were also modified. The percentage of SOC can be used as an indicator to evaluate the overall impact of SOC on the whole OC group.

On page 8 the weighted elemental ratios are discussed. How were these calculated? What is the weighting based on?

**Response:**

The weighting is based on relative intensity. We have changed the part to "relative

intensity weight." in line 159 and 161.

On page 11 it is noted that "In the (+) ESI mode, the CHO group is favorably detected as carbonyl and ester compounds". I would encourage caution with assigning functional groups based on the ESI mode. It is true that these groups can be seen in (+) mode ESI, but so can any group that can solvate the positive charge. Thus, you should also be able to see things like carboxylic acids and alcohols in (+) mode ESI.

**Response:**

We appreciate this comment. Indeed, esters and ketones can be detected in +ESI mode. Our target compounds, however, commonly contained multiple oxygen atoms that is likely in -COOH or -OH functional groups. Indeed, it is inappropriate to name those complex compounds as ester for ketone. What we can confirm is that there is at least one or even more functional groups like ester or ketones.

For a full description, we changed the sentence to lines 185-187:

"In the (+) ESI mode, the CHO groups are favorably detected as carbonyl or ketones with - CHO functional groups, but we can not simply assign those compounds as carbonyl and ester compounds because it was highly likely to contain -COOH or -OH in one molecule. "

On page 11 a comparison is made between the locations in the O/C and H/C Van Krevelen space the ions are observed and the locations for hydrocarbons, lipids, lignin, etc. It is stated that "Saturated hydrocarbons, … and carbohydrates were detected in CHO". I recommend more careful language here as you have not positively identified these chemicals as belonging to these classifications. What you have done is measure molecular formula that have the same O/C and H/C ranges as chemicals that are found in these groups.

**Response:**

We totally agree to this comment. We certainly can not simply assign those compounds as "Saturated hydrocarbons, … and carbohydrates." The classification in VK diagram

is widely used in previous studies. The classification is more used as references to tell audiences what kind of molecules and their environmental impacts. The criteria of the classification is also used in the work of    Bianco et al., 2018.

We have reworded the sentence to lines 192-195:

"We used Van Krevelen diagrams and the criteria from Antony et al. 2014. The classification of Saturated hydrocarbons, unsaturated hydrocarbons, lipids, lignin, and polyphenols is only used as a reference for a better understanding of the distribution of organic molecules and the corresponding environmental impacts."

References

Bianco, A., Deguillaume, L., Vaitilingom, M., Nicol, E., Baray, J.L., Chaumerliac, N., Bridoux, M., 2018. Molecular Characterization of Cloud Water Samples Collected at the Puy de Dome (France) by Fourier Transform Ion Cyclotron Resonance Mass Spectrometry. Environ. Sci. Technol. 52(18), 10275-10285.

Throughout this text comparisons are made between the numbers of different chemical groups that are observed in different samples. However, FT-ICR MS with direct ESI is a non-quantitative technique with large matrix effects. Please include a statement somewhere in the text that acknowledges this and clarifies the level of uncertainty that can be expected.

**Response:**

We totally agree. We have added the statement in the conclusion as a discussion of the limitations of the study (Section 4):

"This work is an attempt to evaluate the evolution of high molecular weight organics using FT-ICR MS. Since the technique is a non-quantitative, and the matrix effect cannot be ignored. Those facts lead to considerable uncertainties."

On page 13 it is noted that "Figure S4 implies that the highest carbon oxidation state (OSc) occurred in compounds with a number of carbon atoms around 20". This is not what I observe in the figure, please modify this text.

**Response:**

We double-checked the Figure, and we have changed the "20" to "15". The changes were also made in the revision (line 211).

On page 14: "CHO with 20 carbon atoms was observed with more oxygen in the Morning and Afternoon samples than that in MEM and Night". I do not observe this in Figure 3. If this difference is present, I suggest including numbers or a table in the supplemental because it is not visibly obvious in the figure.

**Response:**

We have added a table in an Excel file containing those data in supplementary, please see the attachment.

Page 14: "Then, at Night, the highly oxygenated compounds (O number >6, DBE > 5, and carbon atoms >50)…" Why are these called highly oxygenated? That is terminology that is used in atmospheric chemistry for compounds with significantly higher O/C values than what is observed here.

**Response:**

Yes, at this part, the description might be misleading. In the original version, we want to say the organic compound containing more oxygen atoms was found.

We have changed the sentences to lines 220-221:

"Then, at Night, organic compounds containing more oxygen atoms were found (O number >6, DBE>5 and carbon atoms >50) appeared in the nighttime."

For Figure 4, it is difficult to read and interpret this and the other KMD figures. This will be especially true for readers who are red/green colorblind. I recommend changing how this data is presented so that differences between the samples are more observable.

For the KMD analysis comparing CO and COO it is stated that these correspond to the formation of carbonyl and carboxylic acids. However, the formation of carbonyls and carboxylic acids through oxidation reactions does not involve the simple addition of

CO or COO, there are losses of other atoms. It is fine to use these as KMD basis, but the chemical reactions that are underlying these trends are not correctly discussed in this manuscript.

**Response:**

Thanks for the heads up. The color scheme that we used there are friendly to the readers who are red/green colorblind because we have them checked as the request of the ACP journal. We totally agree with the reviewer about the chemical reactions when forming -CO or COO groups. The following statement has been added to the text when discussing the limitations of our work (Section 4):

"When discussing the evolution of HMWOCs, it is noticeable that the formation of higher oxidative groups, such as carbonyl or carboxylic acid groups, commonly were accompanied by loss of other atoms such as hydrogen or even breaking up of the carbon chain. However, those processes are difficult to be illustrated using the FT-ICR MS method."

On page 15: "Those values were recommended for biomass-burning OA in (+) ESI mode (Song et al., 2022)." What does this statement mean and how does it relate to the results listed immediately above?

**Response:**

Sorry for the confusion. We have wanted to stated that those groups were most likely from biomass-burning emissions. And in previous studies, HMWOCs with those parameters were attributed as being from biomass burning.

We have reworded the sentence to lines 247-248:

"The HMWOCs in the range of above parameters were likely from biomass burning, as reported in a previous study by Song et al., 2022."

Please be careful with assigning functional groups to the molecular formula identified throughout the paper. For example, on page 16 it is stated that the CHONs were unsaturated organonitrates. While that may be true, I don't see any fragmentation

analysis that demonstrates these groups are present. Thus, the text needs to be modified to be more careful of the assignments.

**Response:**

We used the criteria from Tang et al. (2020) and Song et al. (2022) . It might be arbitrary to say that CHONs with H:C values between 1.7 and 2.5 and AI< 0.15 were unsaturated organonitrates. A proper attribution as nitrogen-containing organic compounds would be more appropriate because we did not exclude molecules with only one or two oxygen atoms. Also, the H:C and AI values can be influenced by C=C, C=O, and $ONO_2$, $NO_2$ groups. Therefore, we have changed the statement as (lines 250-253):

These CHONs were nitrogen-containing, and most were with –NO2 or –ONO2, namely unsaturated organonitrate. The details of CHON compounds with higher O/N ratios ($\geq$ 3) are discussed in the following text.

On page 16 a series of changes are listed but these are not clearly visible in Figure 5. I suggest adding a table or some additional figure (possibly in the supplemental) to demonstrate that these statements are true.

**Response:**

The average molecular weight decreased during the daytime, from 418 Da in MEM to 402 in Morning, and 399 in Afternoon, then increased to 405 at Night. $O/C_w$ varied from 0.13 to 0.14, while $H/C_w$ was between 1.74 to 1.82. $O/C_w$ was lower than 0.18, and $H/C_w$ was higher than 1.5 (Table 1).

On page 18 it is stated that "The nighttime chemistry strongly removed the CHON1 CH2-homologues with m.w. >400." I do not see this in the figure.

**Response:**

Sorry for your confusion. Our statement is (lines 282-283):

The nighttime chemistry strongly removed the $CHON_1$ CH2-homologues with m.w. >600 (Figure 7c).

A citation of Figure 7c was added, which might be helpful.

On page 19: "the carboxylic acid functional group can be removed due to the reaction with acids such as HNO3 and H2SO4, or be neutralized by NH3 in the particle phase." I am very confused by this statement. What chemical reactions are occurring between carboxylic acids and nitric or sulfuric acid? Do you have references for this chemistry? How would neutralization of the acid impact the measurements in this case? The only impact I can see is a change in solubility (possibly) but the carboxylic acid should still be present.

**Response:**

We are sorry for the ambiguous interpretation of the result.

1. Reactions between nitric acid and carboxylic acid can form organonitrate. For example, Lim et al. (2016) reported organonitrate formation from the reactions between glyoxal and nitric acid in a chamber study. The reaction occurred on wet aerosols via $HNO_3$ uptake. In this work, we found a similar process can occur on large organic molecules in the aerosol phase.

2. Likewise, carboxylic acid can also react with sulfuric acid to form Organosulfate in the aerosol phase (Darer et al., 2011).

3. Carboxylic acid can react with ammonia to form ammonium salt or reduced N-containing particles like pyridine (Zarzana et al., 2012). This process might not impact the measurement but on the environment because the products commonly can absorb light to alter the optical properties of aerosols.

We have modified the following statement to clarify our purpose on this part (lines 291-295):

"…and the carboxylic functional group can be due to the reaction with acids such as $HNO_3$ and $H_2SO_4$ to form organonitrate and organosulfates or be neutralized by $NH_3$ to form ammonium salt or reduced N-containing compounds in the particle phase (Lim et

al., 2016; Darer et al., 2011; Zarzana et al., 2012). In previous studies, those processes were proven to occur in small carboxylic acids. In this work, we can observe a similar process on HMWOCs"

Page 22 it is noted that the CHON compounds with N greater than 8 occurred in both day and night samples and the relative intensities for CHN compounds between 20 and 50 were enhanced. Where is this shown and how is it consistent with the KMD analysis shown in Figure 9?

**Response:**

I suppose the reviewer meant CHN compounds. The N number and relative intensity of high N-containing compounds were enhanced at night, as we can see from Figure 8. In Figure 9, in different KMD plots, we also overserved the enhancement of reduced nitrogen-containing compounds. Hence, the results of both Figures 8 and 9 were consistent.

On page 23 it is noted that you expect acid/base reactions between ammonia and carbonyls. I believe you mean carboxylic acids here as that was what was probed in Dinar et al., 2008. If you do mean carbonyls, please provide more information because this statement is unclear.

**Response:**

Sorry for the typo. We meat "carboxylic acids," and the statement has been changed (line 354).

For the atmosphetric implications section, but also througout the manusciprt. Previous work has been done looking at ESI (+) of organic aerosols with FT-ICR or Orbitrap analysis. How do the results presented here compare to the trends observed in those prior works?

**Response:**

We are grateful for this comment that can clarify the meaning of our work. First, in atmospheric chemistry studies, the ESI (+) is uncommonly used because most studies

were interested in water-soluble brown carbon aerosols, which were commonly used ESI (-) in FT-ICR MS analysis (Li et al., 2022; Han et al., 2022; Qi et al., 2021; Han et al., 2021). Secondly, in most FT-ICR or Orbitrap studies on related subjects, typically daily samples of $PM_{2.5}$ between different days, the trends between samples were hardly considered, and most of the work focused on a comparison of samples in a daily bias. Thirdly, to our knowledge, we might be advanced in the studies of HMWOCs in $PM_{2.5}$ in a typical daily cycle. All of this reason made our comparison with previous studies difficult.

Minor comments

The capillary voltage on page 6 is negative but this is positive ion mode ESI.

**Response:**

In the positive ESI mode, capillary voltage is negative. Please see the snapshot below.

[Figure]

The color scale on Figure 3 is repeated with very small text in the Night. Sample

**Response:**

We have made changes.

In figure 5, what is the color bar?

**Response:**

The color bar is AI values, we have added it in the revision.

On page 23 it is stated that ammonia and biogenic SOA can form light-absorbing oraganonitrates. The chemicals that are produced through this chemistry are not generally organonitrates, they are organonitrogen compounds. Nitrates are a specific functional group and should not be confused with the nitrogen containing compounds found in things like pyridine.

**Response:**

We have changed the statement to "light-absorbing nitrogen-containing organics."(line 339)

Darer, A. I., Cole-Filipiak, N. C., O'Connor, A. E., and Elrod, M. J.: Formation and stability of atmospherically relevant isoprene-derived organosulfates and organonitrates, Environ Sci Technol, 45, 1895-1902, 2011.

Han, H., Feng, Y., Chen, J., Xie, Q., Chen, S., Sheng, M., Zhong, S., Wei, W., Su, S., and Fu, P.: Acidification impacts on the molecular composition of dissolved organic matter revealed by FT-ICR MS, Sci. Total. Environ., 805, 150284, 10.1016/j.scitotenv.2021.150284, 2022.

Han, L., Kaesler, J., Peng, C., Reemtsma, T., and Lechtenfeld, O. J.: Online Counter Gradient LC-FT-ICR-MS Enables Detection of Highly Polar Natural Organic Matter Fractions, Anal Chem, 93, 1740-1748, 10.1021/acs.analchem.0c04426, 2021.

Li, X., Yu, F., Cao, J., Fu, P., Hua, X., Chen, Q., Li, J., Guan, D., Tripathee, L., Chen, Q., and Wang, Y.: Chromophoric dissolved organic carbon cycle and its molecular compositions and optical properties in precipitation in the Guanzhong basin, China, Sci. Total. Environ., 814, 152775, 10.1016/j.scitotenv.2021.152775, 2022.

Lim, Y. B., Kim, H., Kim, J. Y., and Turpin, B. J.: Photochemical organonitrate formation in wet aerosols, Atmos. Chem. Phys., 16, 12631-12647, 10.5194/acp-16-12631-2016, 2016.

Qi, Y., Ma, C., Chen, S., Ge, J., Hu, Q., Li, S.-L., Volmer, D. A., and Fu, P.: Online Liquid Chromatography and FT-ICR MS Enable Advanced Separation and Profiling of Organosulfates in Dissolved Organic Matter, ACS ES&T Water, 1, 1975-1982, 10.1021/acsestwater.1c00162, 2021.

Song, J., Li, M., Zou, C., Cao, T., Fan, X., Jiang, B., Yu, Z., Jia, W., and Peng, P.: Molecular Characterization of Nitrogen-Containing Compounds in Humic-like Substances Emitted from Biomass Burning and Coal Combustion, Environ. Sci. Technol., 56, 119-130, 10.1021/acs.est.1c04451, 2022.

Tang, J., Li, J., Su, T., Han, Y., Mo, Y., Jiang, H., Cui, M., Jiang, B., Chen, Y., and Tang, J.: Molecular compositions and optical properties of dissolved brown carbon in biomass burning, coal combustion, and vehicle emission aerosols illuminated by excitation–emission matrix spectroscopy and Fourier transform ion cyclotron resonance mass spectrometry analysis, Atmos Chem Phys, 20, 2513-2532, 2020.

Turpin, B. J. and Lim, H.-J.: Species Contributions to PM2.5 Mass Concentrations: Revisiting Common Assumptions for Estimating Organic Mass, Aerosol Sci Tech, 35, 602-610, 10.1080/02786820119445, 2001.

Wang, H., Tian, M., Chen, Y., Shi, G., Liu, Y., Yang, F., Zhang, L., Deng, L., Yu, J., Peng, C. J. A. C., and Physics: Seasonal characteristics, formation mechanisms and source origins of PM2.5 in two megacities in Sichuan Basin, China, 18, 865-881, 10.5194/acp-18-865-2018, 2018.

Zarzana, K. J., De Haan, D. O., Freedman, M. A., Hasenkopf, C. A., and Tolbert, M. A.: Optical Properties of the Products of α-Dicarbonyl and Amine Reactions in Simulated Cloud Droplets, Environ Sci Technol, 46, 4845-4851, 10.1021/es2040152, 2012.

---

## Author Response (AR2)

Dear Editor,

we sincerely thank you very much for handling this manuscript. As the request, we have prepared a point-to-point response to all the questions from two reviewers. We hope this response will be satisfactory for both you and them. Again, thank you very much for your work, and we are very glad to contribute to the ACP journal for now and the future.

Introduction: I suggest making the fact that this is a case study of a small subset of samples very clear right away in the introduction to set the realistic scope of the work from the start. Maybe around line 84, "The molecular-level characterization of HMWOC was explored for a subset of four samples, one representing morning, afternoon, night, and midnight".

Response: accepted and done (Line 83).

Methods: I apologize if I missed this, but were peaks from blank filters subtracted from the sampled filters? I see this was also asked by the other reviewer but I cannot see that it was addressed in the manuscript. I recommend very clearly stating your blank subtraction procedure in the text, as it is very difficult to interpret any of the results without understanding how you accounted for filter blanks or solvent blanks.

Response:

The following statement has been added in Section 2.2 (Lines 119-121)

"Extraction blanks and solvent blanks were prepared and analyzed to check for possible contamination. Contaminated peaks in these blanks were removed from organic aerosol samples."

I thank the authors for including the extra SI figure (S4) to address my question about whether there were only a small # of samples included in the analysis and how representative those samples were. It is nice to see that the bulk composition of the aerosol remains consistent across samples. It does look like you have some nice data

from all 52 samples, from the data you put into Figure S4, so I would have loved to see more of it incorporated in the paper. Just because the bulk composition is similar across all 52 samples, it does not mean that you have the same distribution of species within each ion class (e.g., CHO, CHN, CHON), so I still would have loved to see more samples incorporated to the paper since the data do seem to exist and be somewhat processed already. However, I am fine with the current "case study" framing if only 4 samples can be fully incorporated.

Response:

We sincerely appreciate the understanding of the reviewer. Since we have a large data set which is 52 samples, we designed our research framework to dig the dataset. Later on, we would discuss the whole 52 samples of those high molecular weight compounds with their chemical nature, volatility, and source in different articles. Indeed, we will carefully treat the data quality.

Thanks for including the description of how you included N in your OSc calculations. Assuming only a valence of -3 would still introduce some error, especially for the CHON species that are not organonitrates or nitro groups. For CHON, I suggest adding a brief caveat that mentions the possible range of N valence states that could exist in CHON compounds (e.g., you could have a CHON species that's an amine + a carbonyl, rather than an oxidized N atom). Also, did you mean +3 instead of -3 if you have mostly ONO2 or NO2 groups?

Response:

Indeed, a valence of -3 of nitrogen can cause uncertainties. To our knowledge, the issue is quite complicated. Indeed, a complex amine compound with a carbonyl or carboxylic acid can also be a CHON molecule.

Reviewer response:

Thank you for your revisions and your comments. I recommend the following minor changes before acceptance.

1. Thank you for your comments and for the peak list. I understand many of your points in the response, however, I do not see corresponding changes to the manuscript that address them. Please revise the manuscript to address these concerns (either the main manuscript or the supplemental). If you feel that a concern does not warrant a change, please provide more justification for that. Specifically, please include information in the manuscript or in the supplemental that addresses:

A. How the isotopes were handled.

B. How the Na+ ions were identified. I am especially interested in how decisions were made to between choices like formula with many nitrogen atoms compared to one that has fewer Nitrogen atoms and a Na+ ion as an adduct.

C. How the blanks were handled.

D. What fraction of the peaks had a molecular formula assigned to them.

For this last point, I am asking what fraction of the total peak list you were able to assign molecular formulas to. This is often not 100% and it is helpful to know for future researchers because the list of peaks that is provided is often just the identified formulas.

Response:

A:

"The $^{13}C$, $^{18}O$ and $^{15}N$ isotopic molecules were also assigned and analyzed when molecular formula assignment was performed." has been added in *supportive information (Line 28).*

B:

In the raw spectra, we manually compared the mass deviations of the Na-containing and H-containing peaks corresponding to the same molecular formula, which are very close to each other. In addition, we combined the distribution characteristics of the isotopic peaks to distinguish between many nitrogen atoms compared to one that has

fewer Nitrogen atoms and a Na+ ion as an adduct.

"Considering that the number and relative abundance of [M+Na]$^+$ ions are lower than those of [M+H]$^+$ ions , only [M+H]+ was counted and analyzed." has been added in *supportive information (Line 30).*

C:

"Extraction blanks and solvent blanks were prepared and analyzed to check for possible contamination. Contaminated peaks in these blanks were removed from organic aerosol samples." has been added in Section 2.2 (Line 120)

D:

About 50% of mass peaks had a molecular formula assigned to them.

2. I am following up on the following comment in the response to review document:

On page 19: "the carboxylic acid functional group can be removed due to the reaction with acids such as HNO3 and H2SO4, or be neutralized by NH3 in the particle phase." I am very confused by this statement. What chemical reactions are occurring between carboxylic acids and nitric or sulfuric acid? Do you have references for this chemistry? How would neutralization of the acid impact the measurements in this case? The only impact I can see is a change in solubility (possibly) but the carboxylic acid should still be present.

Author's Response:

We are sorry for the ambiguous interpretation of the result.

1. Reactions between nitric acid and carboxylic acid can form organonitrate. For example, Lim et al. (2016) reported organonitrate formation from the reactions between glyoxal and nitric acid in a chamber study. The reaction occurred on wet aerosols via HNO3 In this work, we found a similar process can occur on large organic molecules in the aerosol phase.

2. Likewise, carboxylic acid can also react with sulfuric acid to form Organosulfate in the aerosol phase (Darer et al., 2011).

3. Carboxylic acid can react with ammonia to form ammonium salt or reduced N-containing particles like pyridine (Zarzana et al., 2012). This process might not impact the measurement but on the environment because the products commonly can absorb light to alter the optical properties of aerosols.

We have modified the following statement to clarify our purpose on this part (lines 291-295):

"…and the carboxylic functional group can be due to the reaction with acids such as HNO3 and H2SO4 to form organonitrate and organosulfates or be neutralized by NH3 to form ammonium salt or reduced N-containing compounds in the particle phase (Lim et al., 2016; Darer et al., 2011; Zarzana et al., 2012). In previous studies, those processes were proven to occur in small carboxylic acids. In this work, we can observe a similar process on HMWOCs"

My new Reviewer's response: I recommend being more careful with the language here. I do not think the references that you are using are specifically showing this as a reaction that can occur with carboxylic acids. I do not disagree that this reaction may be possible, but saying that "those processes were proven to occur in small carboxylic acids" needs to be re-written if you are using those references.

Response:

we appreciate the caution of the reviewers. The description in this part was inappropriate. The formation of organonitrates and organosulfate was mainly from the reaction between HNO3 and H2SO4 with alcohols or carbonyls. Carboxylic acid can be removed by reacting with free radicals, ammonia, or other base substances. Therefore, we have revised the part to:

The sentence has been revised to (Lines 292-294):

"…The sinking of carboxylic acids can be done by reacting with free radicals, ammonia, or other base substances to form oligomers, other carbonyl acids, reduced nitrogen-containing compounds, or salts (Ervens et al., 2011).

References

Ervens, B., Turpin, B.J., Weber, R.J., 2011. Secondary organic aerosol formation in cloud droplets and aqueous particles (aqSOA): a review of laboratory, field and model studies. Atmos. Chem. Phys. **11**, 11069-11102. https://doi.org/10.5194/acp-11-11069-2011.